# Functional Identification and Regulatory Active Site Screening of the *DfDXS* Gene of *Dryopteris fragrans*

**DOI:** 10.3390/plants13182647

**Published:** 2024-09-21

**Authors:** Hanxu Zhao, Jiameng Su, Zhaoxuan Zhong, Tongyou Xiong, Weicong Dai, Dongrui Zhang, Ying Chang

**Affiliations:** College of Life Sciences, Northeast Agricultural University, Harbin 150030, China; 18236151062@163.com (H.Z.); sujiameng2022@163.com (J.S.); 18846820987@163.com (Z.Z.); 17671357825@163.com (T.X.); 15084612031@163.com (W.D.); dongruizhang96@gmail.com (D.Z.)

**Keywords:** *Dryopteris fragrans* (L.) Schott, *DfDXS*, gene function, subcellular localization, over-expression of genes

## Abstract

*Dryopteris fragrans* (L.) Schott has anti-inflammatory and antioxidant properties, and terpenoids are important components of its active constituents. The methyl-D-erythritol 4-phosphate (MEP) pathway is one of the major pathways for the synthesis of terpene precursors in plants, and 1-deoxy-D-xylulose-5-phosphate synthase (DXS) is the first rate-limiting enzyme in this pathway. *DXS* has been shown to be associated with increased stress tolerance in plants. In this experiment, two *DXS* genes were extracted from the *D. fragrans* transcriptome and named *DfDXS1* and *DfDXS2*. Based on phylogenetic tree and conserved motif analyses, DXS was shown to be highly conserved evolutionarily and its localization to chloroplasts was determined by subcellular localization. Prokaryotic expression results showed that the number and growth status of recombinant colonies were better than the control under 400 mM NaCl salt stress and 800 mM mannitol-simulated drought stress. In addition, the *DfDXS1* and *DfDXS2* transgenic tobacco plants showed improved resistance to drought and salt stress. *DfDXS1* and *DfDXS2* responded strongly to methyl jasmonate (MeJA) and PEG-mimicked drought stress following exogenous hormone and abiotic stress treatments of *D. fragrans*. The transcriptional active sites were investigated by dual luciferase and GUS staining assays, and the results showed that the STRE element (AGGGG), the ABRE element (ACGTGGC), and the MYC element (CATTTG) were the important transcriptional active sites in the promoters of the two *DXS* genes, which were closely associated with hormone response and abiotic stress. These results suggest that the *DfDXS* gene of *D. fragrans* plays an important role in hormone signaling and response to stress. This study provides a reference for analyzing the molecular mechanisms of stress tolerance in *D. fragrans*.

## 1. Introduction

Terpenoids represent a large class of plant-derived secondary metabolites with five-carbon isoprene (C5H8)n units as their primary structural component [1]. The biosynthesis of terpenoids comprises two distinct pathways: the mevalonate pathway (MVA) located in the cytosol and the methyl-D-erythritol-4-phosphate pathway (MEP) situated within the plastid (Figure 1) [2]. The MVA pathway is responsible for the synthesis of sesquiterpenes and triterpenes, whereas the MEP pathway is primarily involved in the synthesis of monoterpenes, diterpenes, and tetraterpene [3,4]. The intricate network of terpenoid biosynthesis gives rise to the vast array of compounds and their distinctive functions, which are instrumental in conferring plant resilience to environmental stressors, among other biological processes [5]. For example, *Populus trichocarpa* stops insect attacks and attracts insect natural enemies by producing volatile terpenes (mainly monoterpenes and sesquiterpenes) [6,7]. Under adversity stress, plants form a series of response mechanisms to adapt to the changes occurring in the environment, the expression of relevant genes and the regulation of metabolic pathways will be altered, and metabolite types and contents will be changed. *Quercus ilex* L. synthesized about 71% of the total secondary metabolites under drought conditions, and the content of alkaloids and terpenoids increased significantly [8]. Under high-temperature stress, pine trees produced more sesquiterpenoids, including β-butene and α-bergamottin [9].

In the MEP pathway, 1-deoxy-D-xylulose-5-phosphate synthase (DXS) represents the initial rate-limiting enzyme, regulating the supply of deoxy-D-xylulose 5-phosphate (DXP) and consequently influencing terpenoid production in plants [10]. DXS plays a pivotal role in the biosynthesis of DXP from pyruvate and glyceraldehyde 3-phosphate (G-3-P), utilizing thiamine diphosphate (TPP) as a cofactor [11]. Then, 1-deoxy-D-xylulose-5-phosphate reductoisomerase (DXR) catalyses the conversion of 1-deoxy-D-xylulose-5-phosphate to 2-C-methyl-D-erythritol 4-phosphate (MEP), which requires NADPH as a cofactor and Mg^2+^ or Mn^2+^ [12]. The study of *DXS* has revealed the existence of multiple gene families that are both tissue-expression-specific and functionally specific [13].

The first plant *DXS* genes were cloned from *Arabidopsis thaliana* and *Mentha piperita*, followed by *Solanum lycopersicum*, *Pinus massoniana*, *Rosa rugosa*, and *Pelargonium* spp. [13]. The activity of the DXS enzyme can directly influence the quantity of precursors necessary for subsequent terpene synthesis. Additionally, the overexpression of the *DXS* gene exerts significant effects on the downstream products and aspects of plant stress tolerance. The content of photosynthetic pigments, including chlorophyll, was markedly elevated in transgenic tobacco lines that overexpressed *GmDXS* of *Glycine max* (L.) Merr [14]. The expression of the *PmDXS* gene was found to be significantly elevated in drought-treated live seedlings of *Pinus massoniana* Lamb [15]. Transfection of the bacterial *DXS* gene into potato results in longer tubers and premature flowering, suggesting that *DXS* is involved in phenotypic regulation [16]. *PrDXS* in *Pelargonium roseum* Willd was expressed at the highest level in leaves and the least in root tissues, and the relative expression of the *PrDXS* gene decreased as the leaves matured [17], suggesting that the expression level of the *DXS* gene in the plant is tissue-specific.

*Dryopteris fragrans* (L.) Schott is located in Filicales in the taxonomic system and belongs to the *Dryopteris* genus of the Dryopteridaceae family [18], and it is a perennial herbaceous fern found growing on lava tables or in lava crevices formed after volcanic eruptions [19]. The current research on the active ingredients of *D. fragrans* is focused on the extraction and isolation of compounds, pharmacological activity analysis, and gene function exploration. Among them, the active constituents of *D. fragrans* are mainly terpenoids, flavonoids, and meroterpenoids. For example, Hideyuki Ito et al. isolated and identified a variety of compounds in *D. fragrans*, including dryofragin and albicanol [20]. Resorcinol and resorcinol glycosides have also been isolated from *D. fragrans* [21,22]. In addition, gene function, particularly metabolism-related genes, has been studied extensively in *D. fragrans*. For example, Gao [23] and Zhang [24] cloned the terpene synthase (*SQS*) and farnesyl pyrophosphate synthase (*FPPS*) genes from *D. fragrans*, important synthases in the pathway, and laid the foundation for elucidating the terpene pathway in ferns.

In this study, two *DXS* genes were extracted from the *D. fragrans* transcriptome and their encoded proteins were bioinformatically analyzed; the effects of *DXS* in *D. fragrans* on plant stress tolerance were analyzed by prokaryotic expression and tobacco overexpression; the expression pattern of *DfDXS* in *D. fragrans* in response to different hormones and stress treatments; and transcriptional active site analyses were carried out by cloning the promoter sequence of the *DfDXS* gene. These results revealed the function of the *DfDXS* gene in *D. fragrans*, providing a reference for the analysis of the stress tolerance mechanism of *D. fragrans* and facilitating the discovery and use of stress tolerance gene resources.

## 2. Results

### 2.1. Phylogenetic Evolutionary Tree and Conserved Motif Analysis of DfDXSs

In order to understand the evolutionary relationship of the DfDXSs, the amino acid sequences of DfDXS1 and DfDXS2 from *D. fragrans* were homologized by Blastp, and known DXS family sequences from *A. thaliana*, *Daucus carota* and *G. biloba*, as well as ferns such as *Ceratopteris thalictroides* and *Adiantum capillus-veneris*, were downloaded. The different DXS of the ferns are indicated by “−1/2/3”. Phylogenetic analysis was performed using MEGA11.0 software (Figure 2A). DXS from ferns and seed plants were grouped into two large clustering branches, while DXS sequences in seed plants were further grouped into three smaller branches, DXS I, DXS II, and DXS III. In the fern branch, *D. fragrans* DfDXS1 and DfDXS2 are more closely related to *A. capillus-veneris* AcDXS.

Motif prediction analysis of the above DXS amino acid sequences on the MEME online website yielded a total of 10 motifs (Appendix A). Motif1 contains a DXS pyrimidine binding site (DXS_PYR), Motif2 contains a transketolase pyrimidine binding site (Transket_PYR), and both Motif7 and Motif10 contain a transketolase C-terminal binding site (Transketolase_C). The remaining motifs all belong to the deoxyxylulose-5 phosphate synthase family (IPR005477). Furthermore, among the ferns, with the exception of *Ceratopteris thalictroides* CrDXS-1 and *Salvinia natans* SnDXS, which lacked some motifs, the other ferns contained essentially the same motifs as the seed plants. In seed plants, AtDXS3 lacks motifs 4, 8, and 10, and ZmDXS3 lacks motifs 2, 6, 9, 8, and 10. Although the DXS sequences are evolutionarily conserved, the results of the motif analysis suggest that the absence of some motifs may be related to the different functions of the corresponding DXSs.

To further understand the evolutionary relationship of DXS, AcDXS-1, and AcDXS-2 from *A. capillus-veneris*, AtDXS1 and AtDXS2 from *A. thaliana* were selected to perform multiple sequence comparisons with DfDXS1 and DfDXS2 from *D. fragrans* (Figure 2B). The results show that these DXS proteins are very similar, suggesting that DXS is evolutionarily highly conserved. Structural domain analysis revealed that DfDXS1 and DfDXS2 proteins possess a thiamine pyrophosphate binding site (TPP_DXS), an N-terminal transketolase domain (Transket_pyr_3), and a C-terminal transketolase domain (Transketolase_C).

### 2.2. Subcellular Localisation Results for DfDXSs

To further understand the biological function of DfDXS and to explore the site of its function, pCAMBIA2300-EGFP-*DfDXS1* and pCAMBIA2300-EGFP-*DfDXS2* plasmids were constructed by homologous recombination, and the recombinant vectors were successfully obtained after colony PCR identification and sequencing. The leaves of *Nicotiana benthamiana* were transiently transfected with *Agrobacterium tumefaciens* (LBA4404), and the expression of fluorescent proteins was observed by fluorescence confocal microscopy (Figure 3A). The green fluorescence in *N. benthamiana* leaf cells overlapped with the spontaneous red fluorescence of chloroplasts, indicating that DfDXS1 and DfDXS2 were mainly localized in chloroplasts.

### 2.3. Preliminary Validation of Salt and Drought Resistance of DfDXSs in Escherichia coli

To investigate the resistance of DfDXSs, high salt and drought stress conditions were established to observe the growth status of recombinant bacteria pET28a-*DfDXS1* and pET28a-*DfDXS2* after 9 h of IPTG induction compared to that of control pET-28a (Figure 3B). The growth status of pET28a-*DfDXS1* and pET28a-*DfDXS2* recombinant bacteria cultured overnight on LB solid medium was relatively the same as that of control pET-28a, but the growth rate of the recombinant bacteria decreased significantly after culture with stress treatment. After incubation for about 36 h, it was found that the growth status and colony number of the recombinant bacteria were significantly better than those of the control bacteria under salt and drought stress, and it was speculated that the induced expression of DfDXSs might confer some resistance to the host bacteria.

### 2.4. Response of DfDXS1 and DfDXS2 Transgenic Tobacco to Salt and Drought Stress

To investigate the role of *DfDXSs* in plants under adversity stress, the overexpression of *DfDXS1* and *DfDXS2* transgenic tobacco seedlings treated with different adversity conditions to observe their phenotypic changes was induced. The tobacco was irrigated with 200 mM NaCl solution every 3 d and the growth status of the tobacco was observed after 7 d. Leaves from the same tobacco plant were removed and sorted by leaf position (Figure 4A). The wild-type tobacco leaves were relatively more damaged, showing larger areas of wilting and blackening, whereas the transgenic tobacco leaves were relatively less damaged. The physiological indices were measured on the leaves of tobacco seedlings treated with salt stress, and the leaves of wild-type and transgenic tobacco not treated with stress were sampled as the CK group. POD and CAT had different scavenging effects on the reactive oxygen species (ROS) and H_2_O_2_, and the SOD, POD, and CAT of wild-type and transgenic tobacco leaves were all increased under salt stress (Figure 4B–D). Malondialdehyde (MDA) content can reflect the degree of plant injury, and the increase in MDA activity in wild-type tobacco was more significant than that in transgenic tobacco, indicating that the wild-type tobacco group suffered relatively more severe stress and the degree of lipid peroxidation damage was increased (Figure 4E). The overexpression of *DfDXS* improved the resistance of tobacco to salt stress to some extent.

Under drought stress, the old leaves of wild-type tobacco showed yellowing and wilting as the plants lost water. On the 10th day of drought treatment, the wild-type tobacco was severely stressed, whereas the transgenic tobacco overexpressing *DfDXS1* and *DfDXS2* only showed yellowing and wilting of some of the old leaves, and the growth status was significantly better than that of the wild-type tobacco (Figure 5A). Physiological indices were measured on the leaves of soil-grown tobacco subjected to drought stress. Compared with the untreated CK group, the SOD activity of plants exposed to drought stress increased, and the SOD activity of overexpression plants was higher than that of wild-type tobacco (Figure 5B). The POD enzyme activity of overexpression plants was higher than that of wild-type tobacco under drought stress (Figure 5C), and the CAT activity of *DfDXS2* transgenic tobacco was 1.21 times higher than that of wild-type tobacco (Figure 5D). The MDA changes were obvious after drought stress in the wild type, while the MDA content of *DfDXS1* and *DfDXS2* was lower than that of the wild type (Figure 5E), indicating that the overexpression of DfDXS1 and DfDXS2 improved the ability of tobacco to resist drought to some extent.

### 2.5. Effects of Exogenous Hormones and Abiotic Stress on DfDXS Gene Expression

Leaves of *D. fragrans* were treated with the exogenous hormones salicylic acid (SA), abscisic acid (ABA), methyl jasmonate (MeJA), and ethylene tetrachloride (Eth), and the expression pattern of *DfDXS1/2* was analyzed by qRT-PCR. Under SA treatment (Figure 6A), the expression of *DfDXS1* and *DfDXS2* in *D. fragrans* leaves showed a general trend of increasing and then decreasing, with *DfDXS1* peaking at 1 h and *DfDXS2* at 3 h. Under ABA treatment (Figure 6B), the expression of *DfDXS1* and *DfDXS2* was different, and with increasing treatment time, the change in expression level of *DfDXS1* was highest in relative expression at 1 h, while *DfDXS2* showed a decreasing trend. *DfDXS1* and *DfDXS2* responded strongly to MeJA treatment (Figure 6C), and their relative expression levels showed a tendency to increase and then decrease, and both were significantly higher than the control. It is worth noting that the response of *DfDXS2* to MeJA was higher and later than that of *DfDXS1*, and *DfDXS1* peaked at 1 h of MeJA treatment, which was 6.7 times that of 0 h, whereas *DfDXS2* peaked at 6 h of MeJA treatment, which was 12 times that of the control. Under Eth treatment (Figure 6D), the relative expression of *DfDXS1* was lower than the control at all time points except 3 h. The relative expression of *DfDXS2* was lower than the control at all treatment times in Eth.

The expression of *DfDXS1/2* in leaves of *D. fragrans* was analyzed by qRT-PCR under high temperature (HT), low temperature (LT), PEG and NaCl treatments, and the treatment of 0 h indicated the control. Under high-temperature treatment (Figure 6E), the relative expression levels of *DfDXS1* and *DfDXS2* were highest at 0.5 h, and then decreased. Under low-temperature treatment (Figure 6F), the relative expression levels of *DfDXS1* and *DfDXS2* showed a tendency to increase and then decrease and then increase after a sharp decrease at 1 h. Under PEG-simulated drought treatment (Figure 6G), the relative expression of *DfDXS1* was higher than that of the control at all treatment times except for 3 h and then reached the highest expression level at 12 h. The relative expression of *DfDXS2* was lower at 1 and 3 h and higher than the control at the rest of the treatment times. Under NaCl-modelled salt stress (Figure 6H), the expression level of *DfDXS1* increased slightly at 3 h, and NaCl suppressed the expression of the *DfDXS2* gene.

### 2.6. Analysis of the Transcriptional Activity of the DfDXSs Gene Promoter

The expression of certain genes in plants is often closely linked to external environmental influences, and changes in gene expression levels are dependent on upstream promoter regulation [25]. Plant Care’s predicted promoter cis-acting elements show that the promoter of the *DfDXS1* gene contains a number of basic cis-acting elements, TATA box, and CAAT box, as well as cis-acting elements ERE for ethylene, ABRE for abscisic acid, STRE for stress and MYC, and a key regulator of the jasmonic acid signaling pathway (Figure 7A). The promoters of the *DfDXS2* gene contain STRE, a stress response element, MYC, a key regulator of the jasmonate pathway, and TCA, a cis-acting element for the salicylic acid response (Figure 7B).

To identify the core regulatory regions and cis-acting elements of the *DfDXS1* and *DfDXS2* gene promoters, five different-length *DfDXS1/2* gene promoter fragments were obtained by truncation of the promoter length. The obtained promoter truncated fragments were transiently transformed into tobacco, and the same leaf was injected with full-length Δ1, Δ2, Δ3, and Δ4 of the promoter and set up for three treatments, namely MeJA spraying, salt stress (NaCl) and low-temperature stress (LT), and LUC luminescence was observed in the integrated gel imaging analysis system (Figure 7C). Compared with CK, the fluorescence of *DfDXS1* and *DfDXS2* promoters was enhanced under MeJA and NaCl, which may be related to the distribution of the cis-acting elements STRE (AGGGG) and ABRE (ACGTGGC) (Figure 7A,B). Under low-temperature treatment, all five fragments of the *DfDXS1* promoter showed fluorescence, which might be related to the MYC elements (CATTTG) distributed at both ends of the promoter, and MYC was also present in *Pro_DfDXS2_*-Δ2 and *Pro_DfDXS2_*-Δ4, so the fluorescence intensity was obvious.

To verify the promoter response to MeJA, salt stress, and low temperature, GUS staining was performed on a batch of transiently transformed tobacco (Figure 7D). The depth of GUS staining after different stress treatments could reflect the transcriptional activity of the *DfDXS1/2* promoter to some extent, and the overall activity of the *DfDXS1* promoter was stronger than that of the *DfDXS2* promoter. The results of GUS staining under MeJA treatment were consistent with the position of luciferase luminescence in Figure 7C, and the results of *DfDXS2* under low temperature were consistent with the blue color only under the full length of the promoter. In contrast, the *DfDXS2* promoter response site appeared different under salt stress, and *DfDXS2 Pro_DfDXS2_*-full, *Pro_DfDXS2_*-Δ1, *Pro_DfDXS2_*-Δ2, and *Pro_DfDXS2_*-Δ3 appeared more distinctly blue.

### 2.7. Screening for Potential Regulators of DfDXSs

Upon plant exposure to adversity stress, transcription factors act as key regulators that specifically bind to promoters to regulate gene expression [26]. To screen for possible regulators, a gene co-expression network was first constructed (Figure 8). Different colors indicate that the genes were divided into different co-expression networks according to their expression patterns. By calculating the correlation between the modules and the key enzyme genes of the MEP pathway, it was found that the Magenta module was significantly and positively correlated with *D. fragrans DfDXS1*, whereas the correlation between *DfDXS2* and the other modules was low, and it is possible that *DfDXS1* plays a major biological function. More than 500 genes were included in the Magenta module, from which possible transcription factors were screened based on *DfDXSs* gene functions and important cis-acting elements in promoters, and the associated protein physicochemical properties were analyzed and subcellular localization predicted (Table 1).

Finally, the AlphaFold3 online tools were used to predict the binding sites of the transcription factor interacting with the *DfDXS1* promoter of the three-dimensional (3D) protein structures. The prediction results showed that LG33.587 could bind to the ABRE component (Figure 9A), which belongs to the Bzip class of transcription factors and may be involved in the biotic and abiotic stress responses as well as ABA sensitivity and other signals. LG21.134 can bind to two STRE components separately (Figure 9B,C), and it belongs to the Zinc finger-C2H2 class of transcription factors, which are the most widely distributed in eukaryotic genomes and play an important role in enhancing plant resistance. LG15.289 (Figure 9D) and LG.10.782 (Figure 9E) both bind to MYC components, and they belong to the MYB class of transcription factors, which are widely involved in plant growth, development, and metabolic regulation, such as cellular morphology and pattern construction, regulation of secondary metabolism, and response to biotic and abiotic stresses. Therefore, four transcription factors, LG33.587, LG21.134, LG15.289, and LG.10.782, which may be involved in the regulation of DfDXS in response to adversity, were finally selected.

## 3. Discussion

The *DXS* gene family is highly conserved in evolutionary terms. In this study, we selected the DXS of plants whose genomes have been reported, including ferns, gymnosperms, and seed plants. The phylogenetic tree results showed that the DXS of ferns and seed plants were divided into two branches, among which the DXS of seed plants was divided into three branches, which was consistent with the reported results. The ferns as a whole were divided into two branches, of which the closest relative to *D. fragrans* was AcDXS of *A. capillus-veneris*. The phylogenetic analysis of DXS showed that it existed in only one gene copy in some algae and eubacteria, whereas it extended to three subfamilies in land plants [27]. Conserved motif analyses showed that most DXS1 and DXS2 sequences contain almost identical motifs, but DXS3 is usually missing part of the motif, which may be related to the different functions of DXS. The results of multiple sequence comparisons showed high sequence similarity, which may also indicate that DXS is highly conserved evolutionarily.

The subcellular localization of proteins influences their function. We constructed a DfDXS1/2 and EGFP fusion expression vector for transient transformation of tobacco and observed under fluorescence microscopy that the green fluorescence of EGFP coincided with the position of spontaneous red fluorescence in the chloroplasts of tobacco leaves, and the results showed that it was localized in the chloroplasts. The prokaryotic expression vector pET-28a-DfDXS1/2 was constructed and transformed into *E. coli* BL21 Star (DE3) receptor cells, which successfully induced the expression of DfDXS1 and DfDXS2 proteins and initially investigated the role of the DfDXS gene in stress tolerance. The recombinant bacteria induced by IPTG for 9 h and the empty vector control were cultured in salt stress and drought stress medium for 16~32 h. It was found that the recombinant bacteria grew faster than the CK group and the colonies were in a better growth condition, indicating that the expression of DfDXS in *E. coli* could improve its resistance, which provided a reference for the subsequent study of the resistance-related ability of DfDXS in plants. In the future, we will purify and concentrate the protein obtained and carry out in vitro enzyme activity and other related experiments.

To investigate the functions of *DfDXS1* and *DfDXS2* in plants, they were overexpressed in tobacco, and *DfDXS1* and *DfDXS2* transgenic tobacco seedlings were subjected to salt and drought stress. Salt stress affects seed germination, growth, development, flowering, and fruiting processes in plants [28]. The leaves of the WT group were relatively more damaged under salt stress, showing large areas of wilting and blackening, whereas the transgenic tobacco leaves were damaged in relatively smaller areas. The plant response to salt stress requires the regulation of several hormones, among which ABA plays an important role, and osmotic stress leads to the accumulation of ABA in roots and leaves [29]. In the analysis of the *DfDXS* expression pattern, the relative level of the *DfDXS* gene was responsive to both ABA and NaCl, and it was therefore hypothesized that the enhancement of tobacco resistance to salt stress by *DfDXS* correlates with the level of the ABA hormone. Under drought stress, all wild-type old leaves showed yellowing and wilting, whereas transgenic tobacco plants overexpressing *DfDXS1* and *DfDXS2* showed wilting and yellowing only in some old leaves, and their growth status was significantly better than that of WT. In addition, carotenoids and tocopherols produced by the MEP pathway under drought conditions have been shown to play an important role in protection against drought and other related oxidative stresses [30,31]. When plants are exposed to stress, a large amount of reactive oxygen radicals (RORs) are generated in the body, and the protective enzymes SOD, POD, and CAT have the ability to synergistically scavenge RORs to maintain the balance between ROR generation and scavenging and protect the plants from the damage caused by stress. The enzyme activities of SOD, POD, and CAT of the transgenic plants after stress treatment were higher than those of the wild type, indicating that overexpression of *DfDXS* can indirectly improve the stress tolerance of tobacco. The end product of membrane lipid peroxidation is malondialdehyde (MDA), the level of which can be used to reflect the severity of plant stress. The MDA content of wild-type tobacco was relatively higher after stress, indicating that it was more damaged.

SA is an important signaling molecule involved in regulating the biosynthesis of plant secondary metabolites [32]. Under SA treatment, the relative expression of *DfDXS1* and *DfDXS2* in the scented scaly fern peaked at 1 h and 3 h, respectively, and then decreased. ABA had a certain inhibitory effect on the expression of *DfDXS1* and *DfDXS2* genes, but ABA treatment of hairy roots of *Salvia miltiorrhiza* Bunge was able to increase the transcript level of *DXS* [33], suggesting that there is functional variability of the *DXS* gene in different parts of different plants. MeJA was able to increase the levels of terpenoids, such as artemisinin, and the expression of related genes. In this study, we found that *DfDXS* responded strongly to MeJA treatment and its relative expression level was significantly higher than that of the control, which was similar to the expression pattern of *GbDXS* in *Ginkgo biloba* to MeJA [34]. Under abiotic stress treatments, high and low temperatures were able to up-regulate *DfDXS* expression; The relative expression of *DfDXS* in *D. fragrans* was generally up-regulated under PEG-simulated drought stress, which was opposite to the trend of *SlDXS* expression in *Solanum chilense* (Dunal) Reiche under water deficit [35], and it is speculated that this may be related to the specific habitat of *D. fragrans*, which has enhanced tolerance to drought under long-term evolution.

After a plant has been exposed to adverse stress, transcription factors regulate gene expression by binding specifically to the promoter, making the plant somewhat more resistant to damage under adverse conditions [36]. In this study, we analyzed the major cis-acting element sites on the promoters from the point of view of promoter cloning and preliminarily investigated the resistance mechanism of *DfDXS*. The promoters of *DfDXS1* and *DfDXS2* genes contain several basic cis-acting elements, TATA-box and CAAT-box, which are sites recognized by RNA polymerase II and one of the sites for interaction of trans-acting factors with DNA [37]; CAAT-box is thought to control the frequency of transcription initiation [38]. In addition, *DfDXS1* also has phytohormone-related components, such as ERE for ethylene response, MYC for jasmonic acid signaling pathway, STRE for stress response, etc. The promoter of the *DfDXS2* gene contains MYC for the jasmonic acid signaling pathway, TCA for salicylic acid response, STRE for stress response, etc. Promoters play an important role in transcriptional regulation in plants, and transcription factors bind specifically to the cis-acting elements distributed on the promoter to regulate gene expression [39]. In this study, we found that *DfDXS* of *D. fragrans* was involved in plant stress tolerance, and the transcriptional active sites of the promoter were analyzed by both dual luciferase and GUS staining, and it was found that the promoter was responsive to MeJA, NaCl, and low-temperature treatments, which provided a reference to find out the pathway of the stress tolerance function of *DfDXS*. The presence of elements on the responsive promoter truncation segment, the STRE sequence (AGGGG), ABRE (ACGTGGC), and MYC (CATTTG), deserves to be highlighted. STRE is an important stress response element that binds to the C2H2 zinc finger protein family of stress regulators, Msn2 and Msn4 [40]. ABRE can bind to bZIP-like transcription factors and is closely associated with ABA, drought, and salt stress [41]. MYC elements are involved in environmental adaptation in plants and play important functions in abiotic stress response and response to adverse environments [42]. *A. thaliana*, *AtMYC2* binds specifically to the MYC recognition site and is induced by drought and ABA [43]. In conclusion, the analysis of transcriptional active sites of promoters is important for the study of enhancing plant stress tolerance and lays the foundation for finding transcription factors that regulate the expression of *DfDXS1* and *DfDXS2*.

## 4. Materials and Methods

### 4.1. Plant Materials

The spores came from the sporophyte fronds of the perennial *D. fragrans* in the Sanchi area of Wudalianchi and were identified as *Dryopteris fragrans* by researcher Xianchun Zhang of the Institute of Botany, Chinese Academy of Sciences. The leaves were naturally air-dried and the spores were collected in EP tubes and stored in a refrigerator at 4 °C. Spores were germinated in Knop’s liquid medium followed by incubation on the surface of sterile soil at a temperature of 25 ± 1 °C with a 12-h light/12-h dark period. After sporophyte formation, plants were transferred to soil for cultivation (peat soil: vermiculite = 1:3) and incubated at 20–25 °C, maintaining the relative humidity of the sporophytes at 75–95%. Seeds of *Nicotiana tabacum* L. and *N. benthamiana* were kept in the laboratory and grown as adult plants in the greenhouse. The culture conditions are also charcoal soil: vermiculite = 1:3.

### 4.2. Cloning of DfDXS1 and DfDXS2 Target Genes

The Total RNA was extracted from the whole plant of *D. fragrans* using the Total RNA Extraction Kit for Polysaccharide and Polyphenol Plants (centrifugal column type). The Total RNA was used as a template to synthesize the cDNA for cloning according to the Vazyme cDNA reverse transcription instructions. Based on the *D. fragrans* transcriptome database, a local Blast screening was performed using the tbtools software to first obtain candidate sequences, followed by Blastp comparison on the NCBI website. They were named *DfDXS1* and *DfDXS2* based on the full-length sequence of the genes.

### 4.3. Analysis by Bioinformatics

NCBI’s Blastp tool was used to download and search the database for DXS sequences of other species with similar amino acid sequences to DfDXS. The phylogenetic tree was constructed using MEGA11.0 software, and the phylogenetic relationships between DXS from *D. fragrans* and DXS of other plants were analyzed using the Neighbourjoining algorithm with a bootstrap value of 1000. The conserved motifs of DfDXS1 and DfDXS2 encoding proteins from *D. fragrans* were predicted and analyzed using MEME v5.5.7 online tools. DNAMAN was used for multiple sequence comparisons.

### 4.4. Identification of the Subcellular Localisation of DfDXS1 and DfDXS2

The recombinant vectors 2300-*DfDXS1*-EGFP and 2300-*DfDXS2*-EGFP were constructed from the sequences of the DfDXS1 and DfDXS2 genes and the map of the vector pCAMBIA2300-EGFP and transformed into *A. tumefaciens*. Take *N. benthamiana* in good growth condition for about four weeks and ensure that the soil is moist. *A. tumefaciens* was slowly injected into the tissue spaces on the abaxial surface of the leaves using a sterile 1 mL medical syringe (needle removed), avoiding the leaf veins. The leaves were incubated in the dark for 1 d, then in low light for 1 d, and finally in normal light for 1 d. The epidermis of the injected leaves was torn and observed under a fluorescence microscope.

### 4.5. Prokaryotic-Induced Expression of DfDXS1 and DfDXS2

The pET28a was used as a prokaryotic expression vector and *Bam*H I was selected as the digestion site, followed by the addition of homologous arms and ligation, and transferred into the *E. coli* expression strain BL21 Star (DE3). Positive bacteria were verified and the culture was expanded by adding IPTG to a final concentration of 0.5 mM and induced at 28 °C for 3, 6, and 9 h. The protein to be detected is mixed with 5× loading buffer and heated in boiling water for 5 min to denature the protein. IPTG induction of DfDXS was demonstrated by SDS-PAGE gel electrophoresis, Kaomas Brilliant Blue staining, and destaining. LB solid medium containing 400 mM NaCl and 800 mM mannitol was prepared, recombinant bacteria under optimal induction conditions were selected, gradient concentration dilution was performed on them, pET28a-DE3 (Star) was used as a control, and the difference in expression between *E. coli* containing recombinant plasmid and the control group under stress conditions was observed by the drop plate test.

### 4.6. Agrobacterium-Mediated Genetic Transformation of N. tabacum

Based on the *DfDXS1* and *DfDXS2* gene sequences and the plant overexpression vector pCAMBIA2300 profile, the recombinant plasmids 2300-*DfDXS1* and 2300-*DfDXS2* were constructed and transformed and validated in *A. tumefaciens*. *A. tumefaciens* was prepared and then infected with *N. tabacum* (Shanxi), followed by dark incubation at 22 °C for 2–3 d. Subsequently, decontamination and screening cultures were carried out to obtain pure and transgenic *N. tabacum* (Shanxi), and the seeds were sown in nutrient soils together with wild-type (WT) seeds. After 30–45 d of growth, healthy and strong wild-type and transgenic *N. tabacum* (Shanxi) seedlings were selected for salt stress (200 mM NaCl root irrigation) and natural drought stress treatments, respectively. The salt stress treatment was a 3-day root irrigation treatment with samples taken on the 7th day, and the natural drought stress was 10 d. The treated *N. tabacum* plants were photographed and sampled, and the holes were punched in the *N. tabacum* leaves using a 5 mm diameter punch, avoiding the thick veins, and 0.1 g was weighed in each tube, while the untreated group was sampled as above as a control. A sample of *N. tabacum* was analyzed for catalase (CAT), peroxidase (POD), and malondialdehyde (MDA) content as well as superoxide dismutase (SOD) enzyme activities in three biological replicates.

### 4.7. Expression Pattern Analysis of DfDXS1 and DfDXS2

*D. fragrans* was subjected to hormonal treatment and abiotic stress. In the hormone treatment group, the leaves of *D. fragrans* were sprayed with 100 μmol/L methyl jasmonate (MeJA), 100 μmol/L salicylic acid (SA), 500 μmol/L ethylene tetrachloride (Eth), and 10 μmol/L abscisic acid (ABA), and the control group was sprayed with distilled water. In the abiotic stress treatment group, treatments of 42 °C high temperature (HT), 4 °C low temperature (LT), 20% polyethylene glycol (PEG) root irrigation to simulate drought, and 200 mmol/L NaCl whole plant watering were carried out, respectively, and the control group was incubated at normal temperature and watered with distilled water. The concentrations taken were derived from the pre-test. Samples were collected at 0, 0.5, 1, 3, 6, 12, 24, and 48 h after treatment, and plant samples were cut into small pieces and snap-frozen in liquid nitrogen in a refrigerator at –80 °C, with at least three biological replicates for each sample treatment.

RNA was extracted from the above samples, and the cDNA template for qRT-PCR was obtained using a reverse transcription kit. qRT-PCR-specific primers were designed using Primer Premier 6.0 software, and the internal reference gene was *Df18SRNA*. The 2^−ΔΔCT^ method was used to calculate the relative expression of *DfDXS1* and *DfDXS2* genes, and the method was repeated three times for each sample.

### 4.8. Promoter Cloning of the DXS Gene in D. fragrans

The genomic DNA of *D. fragrans* was extracted using a polysaccharide and polyphenol plant genomic DNA extraction kit after grinding the sporophyte leaves of *D. fragrans* by liquid nitrogen flash freezing. The chromosome step was performed using FPNI-PCR (fusion primer and nested integrated PCR) for sequences upstream of the 5′ end of the CDS sequence of the gene. FPNI-PCR was performed using nine sets of random primers SP1-9 designed [44]. The cis-acting elements distributed on the promoters of the *DfDXS1* and *DfDXS2* were predicted online using Plant Care, and the promoters were truncated to their full length according to the location of the cis-acting elements.

### 4.9. Identification of Potential Transcription Active Sites in Promoters

Based on the sequences of the full length, each truncated part of the *DfDXS1* and *DfDXS2* promoters *Pro_DfDXS1_*-full, *Pro_DfDXS1_*-Δ1, *Pro_DfDXS1_*-Δ2, *Pro_DfDXS1_*-Δ3, *Pro_DfDXS1_*-Δ4, *Pro_DfDXS2_*-full, *Pro_DfDXS2_*-Δ1, *Pro_DfDXS2_*-Δ2, *Pro_DfDXS2_*-Δ3 and *Pro_DfDXS2_*-Δ4 were used for the construction of pGreenII0800-Luc and pBI121-GUS vectors. Agrobacterium resuspension containing full-length *DfDXS1* and *DfDXS2* promoters and their respective truncated segments was injected into the abaxial surface of the same *N. benthamiana* leaf and then incubated in darkness for 1 d, in low light for 1 d, and finally in normal light for about 6–8 h. *N. benthamiana* was then subjected to low temperature (LT) at 4 °C, rooting in 200 mM NaCl, simulated salt stress, and 100 μM NaCl for 2 h. *N. benthamiana* was subjected to 4 °C low temperature (LT), 200 mM NaCl root irrigation to simulate salt stress, and 100 μM MeJA spraying, while *N. benthamiana,* without treatment and sprayed only with distilled water, was used as the control group. The treatment time was 4–6 h, after which the leaves were removed.

For the pGreenII0800-Luc recombinant vector, leaves of *N. benthamiana* were sprayed with D-luciferin solution and LUC luminescence was observed using an integrated gel imaging analysis system. For the pBI121-GUS recombinant vector, GUS staining was performed on leaves of *N. benthamiana*, and the formula and steps for GUS staining were taken from Dongrui Zhang [45].

### 4.10. Screening of DfDXSs for Associated Transcription Factors

*D. fragrans* local transcriptome data (*D. fragrans* sporocarp maturation, sporocarp immaturity, sporocarp abscission, MeJA-treated *D. fragrans*, UV-treated *D. fragrans*, *D. fragrans* sporocarp glandular hairs, deglanded hairy leaves, and whole leaves) were used, as well as the genes encoding the seven key enzymes of the *D. fragrans* MEP pathway (*DfDXS1/2*, *DXR*, *MCT*, *CMK*, *MDS*, *HDS*, and *HDR*) were identified, and a gene co-expression network was constructed by MGCNA [46] with a soft domain value of 16. The relevant parts of the Magenta module related to *DfDXS1* were analyzed using GFAP v3.1 software [47], and the transcription factors related to hormone response or adversity stress were screened. The physicochemical properties of the proteins associated with the transcription factors were analyzed using TBtools v2.119 software, the subcellular localization of the transcription factors was predicted using the Cell-PLoc online website, and the three-dimensional (3D) protein structures of the transcription factors were predicted using AlphaFold3 online tools (https://alphafoldserver.com/about, accessed on 18 September 2024).

## 5. Conclusions

Plant terpenoids are diverse and functionally specific and play important roles in plant growth and development and adaptation to environmental changes. DXS is the first rate-limiting enzyme of the MEP pathway. It has been shown that *DXS* is associated with the enhancement of plant stress tolerance. In this study, two *DXS* genes, *DfDXS1* and *DfDXS2*, were obtained from the *D. fragrans* transcriptome database; phylogenetic tree and conserved motif analyses showed that DXS was highly conserved evolutionarily; its subcellular localization was in chloroplasts; the number of recombinant colonies and their growth status were superior to those of controls under 400 mM NaCl and 800 mM mannitol simulated drought stress; and the transgenic tobacco plants of *DfDXS1* and *DfDXS2* had improved resistance to salt and drought stress compared to wild-type tobacco. The number and growth status of the recombinant colonies were better than those of the control group, and the transgenic tobacco plants of *DfDXS1* and *DfDXS2* had an improved ability to resist salt and drought stress compared to wild-type tobacco. Treatment of *D. fragrans* with exogenous hormones revealed that *DfDXS1* and *DfDXS2* are highly responsive to MeJA and SA signals; *DfDXS1* and *DfDXS2* are highly responsive to high-temperature stress and PEG-modelled drought stress under abiotic stress. The promoters of *DfDXS1* and *DfDXS2* genes were cloned chromosome by chromosome, and STRE (AGGGG), ABRE (ACGTGGC), and MYC (CATTTG) were found to be important cis-acting elements determining the transcriptional activity of the promoters, which were closely related to hormone response and abiotic stress. Finally, tertiary structure prediction of the screened transcription factors related to hormone response or adversity stress was performed using AlphaFold3. In this study, we found that *D. fragrans DfDXS* plays a role in plant resistance to abiotic stress, which provides a basis for the analysis of the resistance mechanism of *D. fragrans* and is conducive to the discovery and exploitation of resistance gene resources.

## Figures and Tables

**Figure 1 plants-13-02647-f001:**
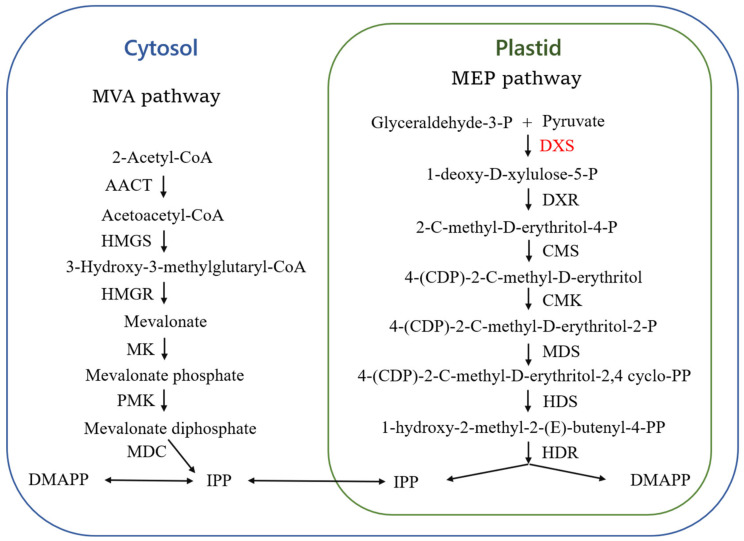
Terpenoid biosynthesis pathway. DXS (marked in red), the first rate-limiting enzyme in the MEP pathway under focus.

**Figure 2 plants-13-02647-f002:**
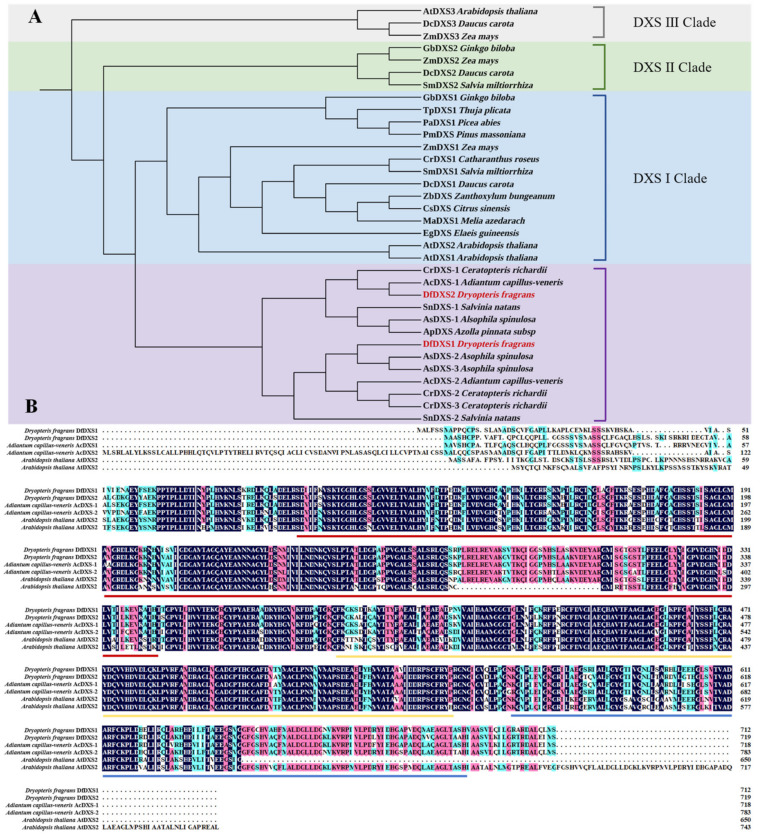
Evolutionary analyses between DfDXS and other species. (**A**) Phylogenetic tree analysis of the DXS gene family. DXS of the ferns, purple; DXS I, blue; DXS II, green; DXS III, grey. (**B**) Multiple sequence alignment of DXS protein. The red underline represents the binding site of thiamine pyrophosphate (TPP_DXS); the yellow underline represents the N-terminal domain of transketolase (Transket_pyr_3); and the blue underline represents the C-terminal domain of transketolase (Transketolase_C).

**Figure 3 plants-13-02647-f003:**
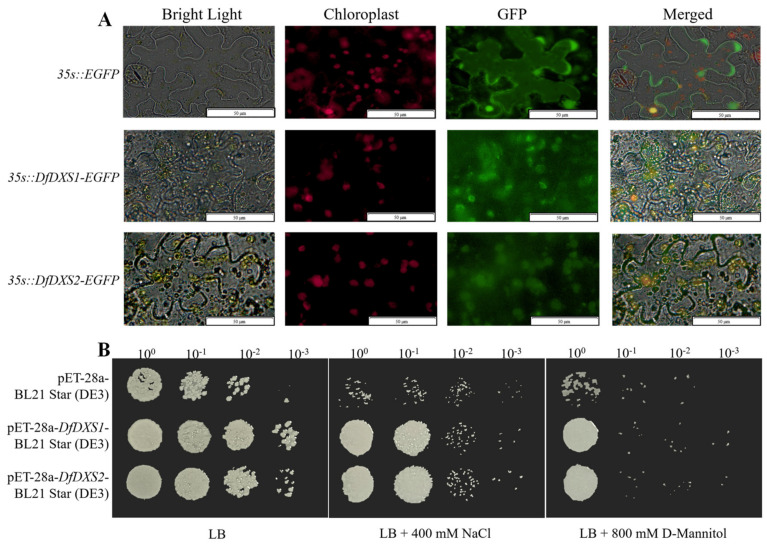
Subcellular localization and prokaryotic expression of DfDXSs. (**A**) Subcellular localization of DfDXS1 and DfDXS2, Bar = 50 μm. (**B**) Growth status of recombinant bacteria and control bacteria under simulated stress, 0^0^~10^−3^ represent the dilution gradients of *E. coli* liquid culture.

**Figure 4 plants-13-02647-f004:**
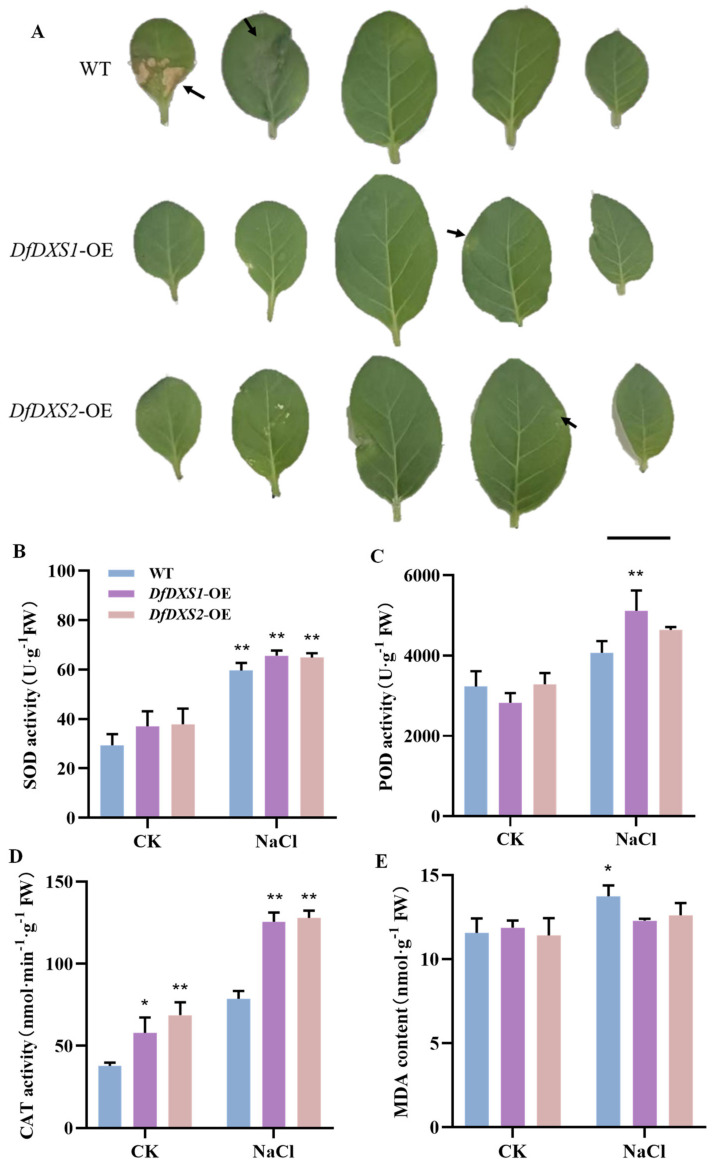
Changes associated with the overexpression of *DfDXS1* and *DfDXS2* in transgenic tobacco under salt stress. (**A**) Comparison of the degree of leaf damage on day 7 after root irrigation treatment with 200 mM NaCl solution. Black arrows indicate the damaged parts of tobacco leaves, bar = 20 mm. (**B**–**E**) Physiological parameters of *DfDXS1/2* overexpressing transgenic tobacco plants under salt stress. An asterisk (*) in the figure indicates that the significant level is 0.05, two asterisks (**) indicate a significant level of 0.01.

**Figure 5 plants-13-02647-f005:**
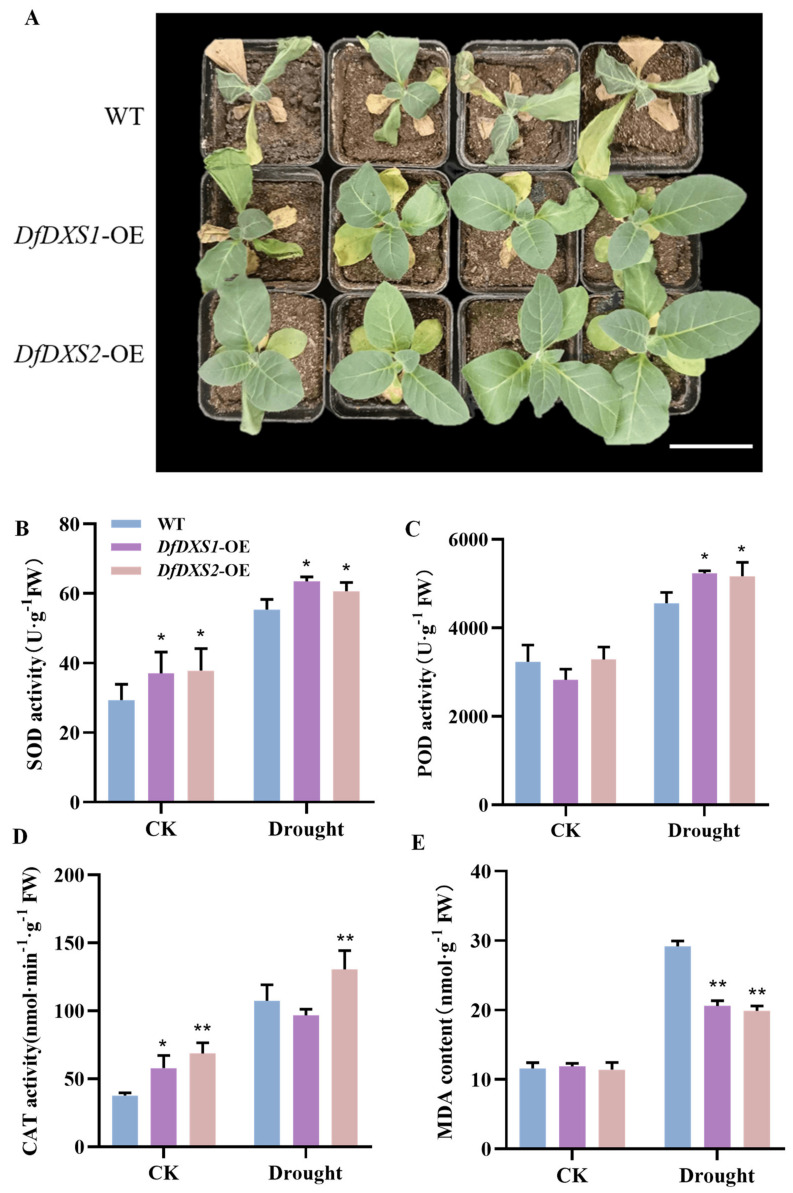
Changes associated with *DfDXS1*, *DfDXS2* transgenic tobacco under drought stress. (**A**) Phenotypic changes in tobacco treated with natural drought for 10 d compared to wild-type tobacco (WT), bar = 7 cm. (**B**–**E**) Physiological indices measured in transgenic tobacco plants overexpressing *DfDXS1*, *DfDXS2* under drought stress. An asterisk (*) in the figure indicates that the significant level is 0.05, two asterisks (**) indicate a significant level of 0.01.

**Figure 6 plants-13-02647-f006:**
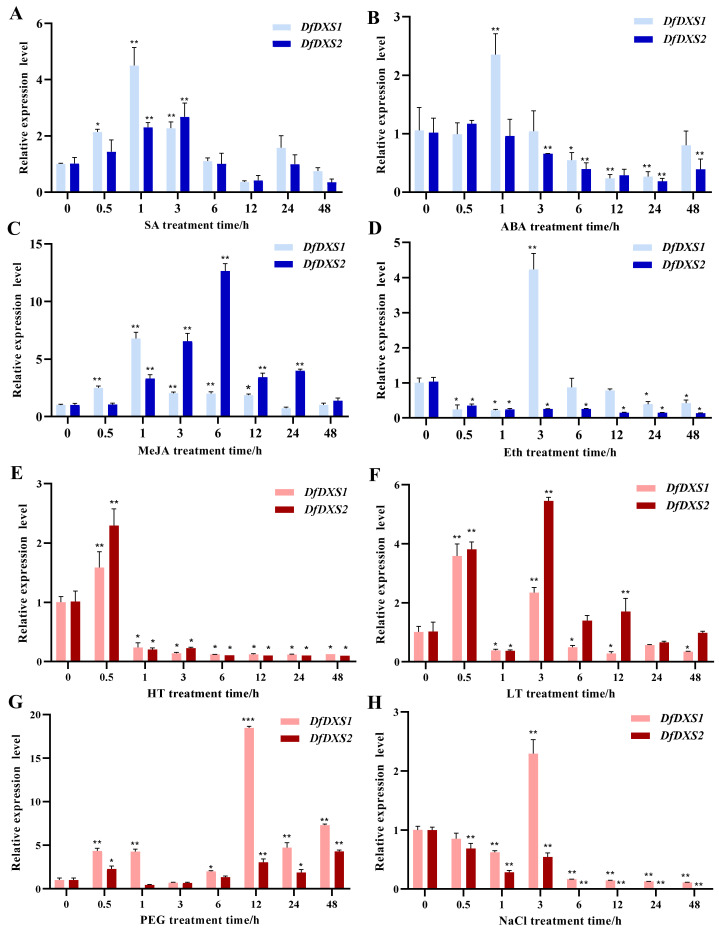
Expression patterns of *DfDXS1* and *DfDXS2* genes under different treatments. (**A**–**D**) Changes in the relative expression levels of *DfDXS1* and *DfDXS2* under different phytohormone treatments. (**E**–**H**) Changes in the relative expression levels of *DfDXS1* and *DfDXS2* under different stress treatments. An asterisk (*) in the figure indicates that the significant level is 0.05, two asterisks (**) indicate a significant level of 0.01, three asterisks (***) indicate that the significant level is 0.001, four asterisks.

**Figure 7 plants-13-02647-f007:**
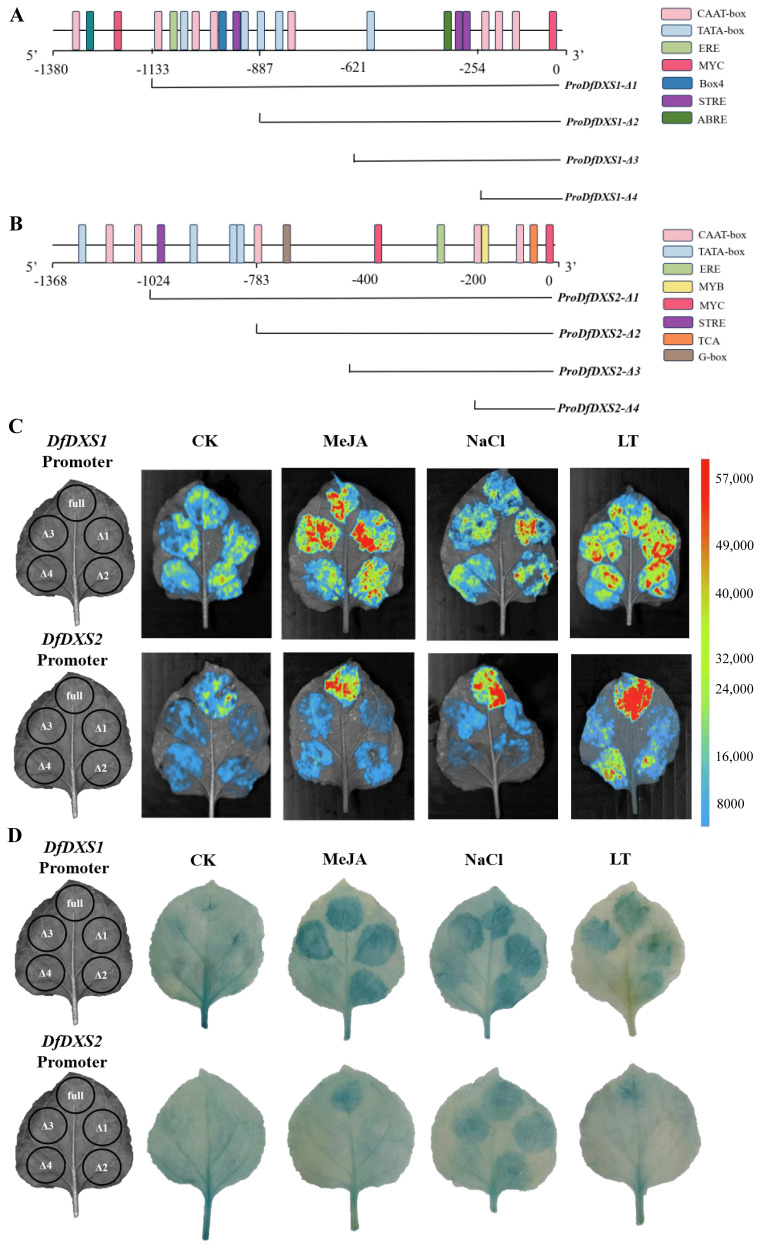
Truncation of the *DfDXS1/2* promoter and analysis of transcriptional activity. (**A**) Cloning of truncated fragments and prediction of cis-acting element distribution of *DfDXS1* gene promoter in *D. fragrans.* (**B**) Cloning of truncated fragments and prediction of cis-acting element distribution of *DfDXS2* gene promoter in *D. fragrans.* (**C**) Fluorescence response of truncated fragments of *DfDXS1/2* promoter of *D. fragrans* under different stresses. (**D**) GUS staining of truncated fragments of the *DfDXS1/2* promoter of *D. fragrans* under different stresses. The left part of C and D shows the positional distribution of the *DfDXS1/2* promoter truncation segments transiently transformed in tobacco leaves, where full denotes *Pro_DfDXS1/2_*-full, Δ1 denotes *Pro_DfDXS1/2_*-Δ1, Δ2 denotes *Pro_DfDXS1/2_*-Δ2, Δ3 denotes *Pro_DfDXS1/2_*-Δ3, and Δ4 denotes *Pro_DfDXS1/2_*-Δ4.

**Figure 8 plants-13-02647-f008:**
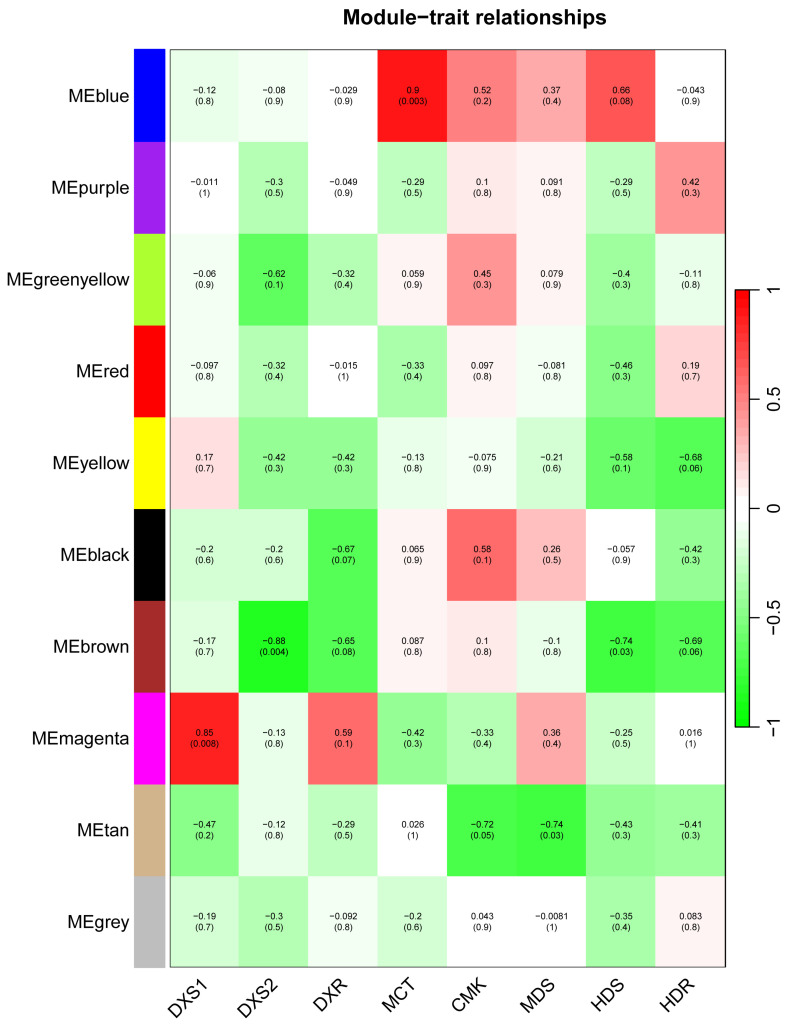
Heatmap of correlation between key enzyme genes of the MEP pathway and different gene modules of the transcriptome. The darker the color, the greater the correlation. DXS, 1-deoxy-D-xylulose-5-phosphate synthase. DXR, 1-Deoxy-d-xylulose-5-phosphate reductoisomerase. MCT, 2-C-methyl-D-erythritol-4-phosphate cytidylyltransferase. CMK, 4-diphosphocytidyl-2-C-methyl-D-erythritol kinase. MDS, 2-C-Methyl-D-erythritol-2,4-cyclodiphosphate synthase. HDS, hydroxide methyl enylamino 4-cyclodiphosphate synthase. HDR, hydroxy-2-methyl-2-(E)-butenyl 4-diphosphate reductase.

**Figure 9 plants-13-02647-f009:**
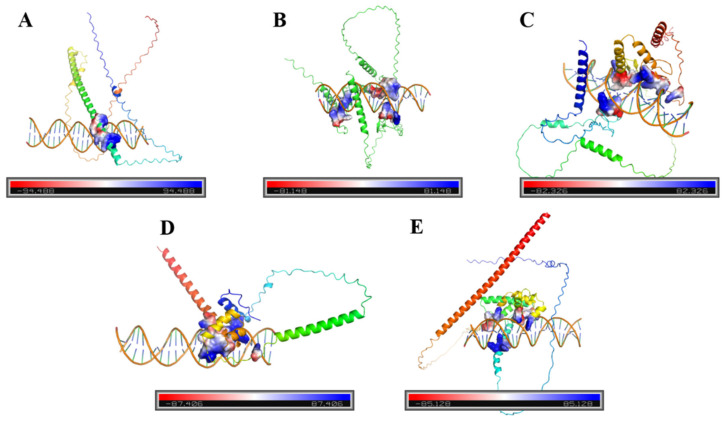
Binding sites and 3D structure prediction of *DfDXS1* cis-acting elements with transcription factors. (**A**) LG33.587 bonded to the ABRE assembly; (**B**,**C**) LG21.134 bonded to each of the two STRE assemblies; (**D**) LG15.289 bonded to the MYC assembly; and (**E**) LG.10.782 bonded to the MYC assembly.

**Table 1 plants-13-02647-t001:** Analysis of the physicochemical properties and prediction of subcellular localization of potential regulators of *DfDXS1*.

Gene ID	Location (bp)	CDS Length (bp)	Annoation	Protein Length (aa)	MW (kDa)	Predicted Location (s)
LG33.587	1069–1074	732	Bzip	269	29,727.31	Nucleus.
LG21.134	1106–1111 1111–1116	993	Zinc finger-C2H2	353	39,596.33	Nucleus.
LG40.463	-	474	Myb DNA-bind 4	180	20,556.24	Nucleus. Peroxisome.
LG22.782	-	717	Myb DNA-bind 4	238	27,224.65	Nucleus.
LG24.556	-	519	Myb DNA-bind 4	192	22,187.27	Nucleus.
LG15.289	1343–1349	1122	Myb DNA-bind 4	393	44,059.22	Nucleus.
LG10.782	1343–1349	567	Myb DNA-binding	208	24,249.76	Nucleus.

## Data Availability

Data is contained within the article or Appendix A.

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
