# Peer review of "Functional Identification and Regulatory Active Site Screening of the *DfDXS* Gene of *Dryopteris fragrans"

_plants, 2024, doi:10.3390/plants13182647_

Round 1

Reviewer 1 Report

Comments and Suggestions for Authors

There are several typing errors of different types on the MN, starting from the sentence For example,Populus trichocarpa in the Introduction

The Introduction contains several arguments without evident relation with the argument of the aims of the MN. It is necessary a careful revision.

The sentence Dryopteris fragrans (L.) Schott belongs to the Dryopteridaceae Dryopteris, a perennial 64 herbaceous fern found growing on lava tables or in lava crevices formed after volcanic 65 eruptions[17] is confusing and must be revised. Dryopteris is the genus and not the species.

As the international rules of publication, the plant species should be identified by an expert in systematics and voucher samples deposited for further controls, as well as the origin. This a crucial argument.

As reported in several sentences, the main goal of the MN is the production of selected plant terpenes. However, the relation of the research on terpenes production , is not clear. Although effects on the biosynthetic pathways can be accepted. Which of the terpenes should be produced in consequence? Which is the relation with the selected plant? Why this species should be preferred? Furthermore, the authors should make evident for the reader the real consequences of the research. In particular in this case, where the authors in several sentences of the MN report the impact of the terpenes in ordinary life.

The final comment is that the MN should modify the MN in accordance with the above considerations, with intent to evidence in a better way the importance and consequence of the research.

Comments on the Quality of English Language

the meaning of several sentences could be changed to help the reader in understanding 

Author Response

Comments 1: There are several typing errors of different types on the MN, starting from the sentence, for example, Populus trichocarpa in the Introduction.

Response 1: Thank you for pointing this out. We agree with this comment. Therefore, we have amended the occurrence of Populus trichocarpa Torr. & Gray. in the article to Populus trichocarpa. We will review the entire article and correct similar typos in the article.

Comments 2: The Introduction contains several arguments without evident relation with the argument of the aims of the MN. It is necessary a careful revision.

Response 2: Thank you for pointing this out and we agree with your comments. We have made revisions to the preface section, including deleting the introduction in the preface on the inflammatory response of macromolecular polysaccharides from Dryopteris fragrans and adding a section on the effects of changes in terpenoid content on plant stress tolerance.

Comments 3: The sentence Dryopteris fragrans (L.) Schott belongs to the Dryopteridaceae Dryopteris, a perennial 64 herbaceous fern found growing on lava tables or in lava crevices formed after volcanic 65 eruptions [17] is confusing and must be revised. Dryopteris is the genus and not the species.

Response 3: Thank you very much for pointing this out and we agree with the comment. We will consult the relevant information and carefully revise the description of Dryopteris fragrans to match the facts.

Comments 4: As the international rules of publication, the plant species should be identified by an expert in systematics and voucher samples deposited for further controls, as well as the origin. This a crucial argument.

Response 4: Thank you for pointing this out and we agree with the comment. We will add a note on the identification and origin of Dryopteris fragrans to the article to comply with international publication rules.

Comments 5: As reported in several sentences, the main goal of the MN is the production of selected plant terpenes. However, the relation of the research on terpenes production, is not clear. Although effects on the biosynthetic pathways can be accepted. Which of the terpenes should be produced in consequence? Which is the relation with the selected plant? Why this species should be preferred? Furthermore, the authors should make evident for the reader the real consequences of the research. In particular in this case, where the authors in several sentences of the MN report the impact of the terpenes in ordinary life.

Response 5: Thank you very much for pointing out the problem. We agree with your suggestion. DXS is the first rate-limiting enzyme of the MEP pathway, which not only influences the synthesis of terpene precursors, but also has been found to be associated with plant resistance to external environmental stresses. Our main objective is to investigate the function of DfDXS in stress resistance, which will be beneficial for the discovery and utilisation of stress-resistant gene resources. The active components in D. fragrans are mainly terpenoids, flavonoids and mesoterpenoids, which have different pharmacological effects in their compounds, such as antibacterial, anti-inflammatory, antioxidant and antitumour effects. Therefore, the study of resistance genes of D. fragrans and enhancement of its resistance can improve the yield of D. fragrans, which can be better applied to production and benefit human beings. We will carefully revise inappropriate statements in the essay to make them more relevant to the central idea we want to convey.

Reviewer 2 Report

Comments and Suggestions for Authors

This manuscript reported the functional investigation of DfDXS gene and its effect on plants stress tolerance. Initially, the relationship between DfDXS gene and stress tolerance was established by the observation of better effect in the DfDXS overexpressed plants compared to the control. Further, the transcriptional active sites were investigated by dual luciferase and GUS staining assays. Overall, it is a plant molecular biology paper, with a focus on gene functionality research. As far as I am concerned, this manuscript can be assigned as minor revision.

Here are my recommendations

1) It is recommended to add a figure to explain the MEP pathway, instead of only using words. It will be much clearer to establish the relationship among terpenoids, DXS, and DfDXS.

2) Any literature reported the effect of terpenoids on plants stress tolerance? If so, please add them into the introduction. In the content, it seems that the relationship between DfDXS and plants stress tolerance was connected, but is it due to the increasing amount of terpenoids?

Author Response

Comments 1: It is recommended to add a figure to explain the MEP pathway, instead of only using words. It will be much clearer to establish the relationship among terpenoids, DXS, and DfDXS.

Response 1: Thank you for raising this point. We agree with your comment. Therefore, we have added a flowchart to the preface section on the MEP pathway (page 2 of the manuscript) to help the reader visualise the relationship between DXS and terpenoids.

Comments 2: Any literature reported the effect of terpenoids on plants stress tolerance? If so, please add them into the introduction. In the content, it seems that the relationship between DfDXS and plants stress tolerance was connected, but is it due to the increasing amount of terpenoids?

Response 2: Thank you for your comments. We have carefully reviewed the information and there are now similar reports showing that changes in terpenoid content can influence plant stress tolerance. We have added these to the preface section, including the fact that Quercus ilex L. showed a significant increase in terpenoid content under drought conditions and that pine trees produce more sesquiterpenes under high temperature stress.

Reviewer 3 Report

Comments and Suggestions for Authors

Why red fonts were used for fig and table?

Line 311: figure 7 is difficult to interpreter without explanation what means MEblue   …. MEmagenta?

The enzymes nicknames could be explained, also

Figure 7: Heat map of correlation…

or Heatmap ?

 Lines 328-329

Liping Zeng et al. found that the phylogenetic analysis of  DXS showed that it existed in only one gene copy in some algae and eubacteria, whereas   it extended to three subfamilies in land plants [23].

Lines: 353,and: 358,366,377,381,386,390,394,406,415,416,417, and  in the Introduction 

plants[24].

Space between word and [ of citation  is needed

Line 423: capital letter  

Line 424 ???

4.1. Plant materials

Sporophytic plants of D. fragrans: Spores were collected from the sporophyte leaves  of the perennial D. fragrans in the Wudalianchi scenic area. The leaves were naturally air-  dried and the spores were collected in EP tubes and stored in a refrigerator at 4°C. Seeds  of Nicotiana tabacum(Shanxi) and N. benthamiana were kept in the laboratory and grown as  adult plants in the greenhouse under the same conditions as Sporophytic plants of D. fragrans.

Better

Dryopteris fragrans (L.) Schot  spores were collected from the sporophyte leaves  of the perennial plant  in the Wudalianchi scenic area. The leaves were naturally air- dried and the spores were collected in EP tubes and stored in a refrigerator at 4°C.

Nicotiana tabacum(Shanxi) and N. benthamiana seeds were kept in the laboratory and grown as   adult plants in the greenhouse under the same conditions as sporophytic plants of D. fragrans.

Line 428

“greenhouse under the same conditions “    I think that  greenhouse parameters during the plants grown must be précised

Lines 460-461, no results of SDS-PAGE !

The proteins to be detected were denatured and subjected to SDS-PAGE gel electrophoresis, stained with Cauldron Brilliant Blue dye and finally destained with a destaining solution to remove the background colour.

Line 482: methods of activities measure?

The enzyme activities of catalase (CAT), peroxidase (POD), superoxide dismutase (SOD) and malondialdehyde (MDA) were measured on the sampled N. tabacum leaves in three biological replicates.

Line 619: why capital letters in the title ?

Ito, H.; Muranaka, T.; Mori, K.; Jin, Z.X.; Yoshida, T. DRYOFRAGIN AND ASPIDIN PB, PISCICIDAL COMPONENTS FROM  DRYOPTERIS FRAGRANS. Chemical & Pharmaceutical Bulletin 1997, 45, 1720-1722.

Line 648

ofGinkgo biloba

Author Response

Comments 1: Why red fonts were used for fig and table?

Response 1: Thank you for pointing this out. The striking red font was previously used to make it easier for editors to see, we have now reworked the manuscript, changed the fig and table to black font and uploaded the new manuscript with the corrections.

Comments 2: Line 311: figure 7 is difficult to interpreter without explanation what means MEblue …. MEmagenta?

The enzymes nicknames could be explained, also

Figure 7: Heat map of correlation…or Heatmap ?

Response 2: Thank you for pointing this out. We agree with this comment. Firstly, Figure 7 is a heatmap of the correlation of key enzyme genes with different gene modules of the transcriptome. MEblue .... MEmagenta represents a collection of individual genes divided according to expression trends, and we will upload a schedule of gene IDs for each module. Second, each enzyme in the figure is in abbreviated form, and we will write the full name of each enzyme in a new manuscript figure note. Finally, we have modified the Heat map in Figure 7 to Heatmap.

Comments 3: Lines 328-329

Liping Zeng et al. found that the phylogenetic analysis of DXS showed that it existed in only one gene copy in some algae and eubacteria, whereas it extended to three subfamilies in land plants [23].

Response 3: Thank you for pointing out the issue with the article's word order, and we agree with this comment. We will therefore carefully review and revise this issue throughout the article.

Comments 4: Lines: 353, and: 358, 366, 377, 381, 386, 390, 394, 406, 415, 416, 417, and in the Introduction plants[24].

Space between word and [ of citation  is needed

Response 4: Thank you very much for pointing out the article formatting issue and we agree with this comment. We will carefully review and revise this issue throughout the article.

Comments 5: Line 423: capital letter

Response 5: Thank you for pointing out the article formatting issue and we agree with this comment. We will carefully review and revise this issue throughout the article.

Comments 6: Line 424 ???

4.1. Plant materials

Sporophytic plants of D. fragrans: Spores were collected from the sporophyte leaves  of the perennial D. fragrans in the Wudalianchi scenic area. The leaves were naturally airdried and the spores were collected in EP tubes and stored in a refrigerator at 4°C. Seeds  of Nicotiana tabacum(Shanxi) and N. benthamiana were kept in the laboratory and grown as  adult plants in the greenhouse under the same conditions as Sporophytic plants of D. fragrans.

Better

Dryopteris fragrans (L.) Schot  spores were collected from the sporophyte leaves  of the perennial plant  in the Wudalianchi scenic area. The leaves were naturally air- dried and the spores were collected in EP tubes and stored in a refrigerator at 4°C.

Nicotiana tabacum(Shanxi) and N. benthamiana seeds were kept in the laboratory and grown as adult plants in the greenhouse under the same conditions as sporophytic plants of D. fragrans.

Response 6: Thank you for pointing out the problem with the article's statements, and we agree with this suggestion. We will therefore correct the original statement and re-upload it.

Comments 7: Line 428

“greenhouse under the same conditions” I think that greenhouse parameters during the plants grown must be précised

Response 7: Thank you very much for pointing this out. We agree with this comment. We will add more details about the greenhouse parameters of the plants during growth in the article.

Comments 8: Lines 460-461, no results of SDS-PAGE !

The proteins to be detected were denatured and subjected to SDS-PAGE gel electrophoresis, stained with Cauldron Brilliant Blue dye and finally destained with a destaining solution to remove the background colour.

Response 8: Thank you for pointing this out. We agree with this comment. We will add the SDS-PAGE results in detail in the article and include the results in a supplementary figure as an indication of the induction of IPTG by DfDXS.

Comments 9: Line 482: methods of activities measure?

The enzyme activities of catalase (CAT), peroxidase (POD), superoxide dismutase (SOD) and malondialdehyde (MDA) were measured on the sampled N. tabacum leaves in three biological replicates.

Response 9: Thank you for pointing this out. We agree with this comment. We will carefully review the order of the sentences in the article to make it easier for readers to understand.

Comments 10: Line 619: why capital letters in the title?

Ito, H.; Muranaka, T.; Mori, K.; Jin, Z.X.; Yoshida, T. DRYOFRAGIN AND ASPIDIN PB, PISCICIDAL COMPONENTS FROM DRYOPTERIS FRAGRANS. Chemical & Pharmaceutical Bulletin 1997, 45, 1720-1722.

Response 10: Thank you very much for pointing this out. We agree with the comment. This was an oversight on our part and we will double check and correct the formatting issues in the references.

Comments 11: Line 648 ofGinkgo biloba

Response 11: Thank you for pointing this out. We agree with the comment. The lack of space between the two words was an oversight on our part. We will double check and correct similar errors in the article.

Round 2

Reviewer 1 Report

Comments and Suggestions for Authors

I am afraid in the MN is not clear the difference between terpenes and terpenoids. The structure of a terpene is made only by C5 units, whereas a terpenoid is made by C5 units and others. It is like human and humanoid, in science fiction or better alkali and alkaloid. To my opinion, the possible current tendency of using the two terms like synonymous is matter of confusion, beside ignorance.

About the sentences The secondary metabolites synthesised under drought conditions accounted for 41 about 71% of the total metabolites, and the contents of alkaloids and terpenoids increased 42 significantly in Quercus ilex L. The metabolites synthesised under drought conditions ac- 43 counted for about 71% of the total metabolites, and the contents of alkaloids and terpe- 44 noids increased significantly [8]. What a repeated sentence, did the authors recheck the MN? In any case, the sentence is disputable and it is not clear if it a general consideration or related to the oaks.

About 73 longs to the Dryopteris genus, the genera must be written in italic.

I have no idea of what is a mesoterpenoid. Perhaps is it a meroterpenoid?

About the sentence For example, Hideyuki Ito et al. isolated and identified 79 a variety of compounds in D. fragrans, including Dryofragin and Albicanol [20]. The reference 20 dos not correspond at all. The Ito’s paper is lacking. Why the names of the compounds in capital?

Our re- 80 search group isolated resorcinol and resorcinol glycosides from D. fragrans. It must be a reference.

Therefore, all the phytochemical information can not be accepted.

Check again the MN, to avoid errors like and incubated at (20-25) °C to be changed in and incubated at 20-25 °C and many others in all the MN.

I suggest to add a list of abbreviations, as usual in this case.

Final comment: there are still too many errors of different types, beside the lack of adequate changes in consideration of previous comments.

Author Response

Comments 1: I am afraid in the MN is not clear the difference between terpenes and terpenoids. The structure of a terpene is made only by C5 units, whereas a terpenoid is made by C5 units and others. It is like human and humanoid, in science fiction or better alkali and alkaloid. To my opinion, the possible current tendency of using the two terms like synonymous is matter of confusion, beside ignorance.

Response 1: Thank you for pointing this out and we agree with you. This should have been a mistake in translation at the time, I am very sorry. This is our oversight; we didn't check it carefully at that time. We will double-check the content of the article again and make changes.

Comments 2: About the sentences “the secondary metabolites synthesised under drought conditions accounted for 41 about 71% of the total metabolites, and the contents of alkaloids and terpenoids increased 42 significantly in Quercus ilex L. The metabolites synthesised under drought conditions ac- 43 counted for about 71% of the total metabolites, and the contents of alkaloids and terpe- 44 noids increased significantly [8].” What a repeated sentence, did the authors recheck the MN? In any case, the sentence is disputable and it is not clear if it a general consideration or related to the oaks.

Response 2: Thank you very much for pointing this out, we agree with you. I'm sorry for the bad reading experience, I was so careless that I didn't even find such an obvious error. I will carefully check the article for similar issues and revise them.

Comments 3: About 73 longs to the Dryopteris genus, the genera must be written in italic.

Response 3: Thank you very much for pointing this out and we agree with the comment. It was an oversight on our part to have overlooked this previously and we have changed the genera to an italicized format.

Comments 4: I have no idea of what is a mesoterpenoid. Perhaps is it a meroterpenoid?

Response 4: Thank you very much for pointing this out and we agree with the review. It was an oversight on our part that led to the spelling error. We will double-check and correct this issue in the article.

Comments 5: About the sentence for example, Hideyuki Ito et al. isolated and identified 79 a variety of compounds in D. fragrans, including Dryofragin and Albicanol [20]. The reference 20 dos not correspond at all. The Ito’s paper is lacking. Why the names of the compounds in capital?

Response 5: Thank you very much for pointing out the problem. We agree with your suggestion. We have previously made changes to the content of the references, which may have led to the mismatch of reference 20. We are very sorry that this was an oversight on our part, and we will double-check and correct this issue again. In addition, we have also checked and changed the names of the compounds in detail.

Comments 6: Our re- 80 search group isolated resorcinol and resorcinol glycosides from D. fragrans. It must be a reference.

Response 6: Thank you very much for pointing out the problem and we agree with you. This was an oversight on our part and we will add the corresponding reference to the part of the sentence described.

Comments 7: Check again the MN, to avoid errors like and incubated at (20-25) °C to be changed in and incubated at 20-25 °C and many others in all the MN.

Response 7: Thank you very much for pointing out the problems and we agree with your comments. We are sorry that we had these formatting issues and we will double check the MN and revise it carefully.

Comments 8: I suggest to add a list of abbreviations, as usual in this case.

Response 8: Thank you very much for your question. I'm sorry I can't see the exact location where you mentioned your comment on this platform, I hope you can describe exactly where it is so I can better change the content of the article.

Comments 9: Final comment: there are still too many errors of different types, beside the lack of adequate changes in consideration of previous comments.

Response 9: Thank you very much for pointing out the problem and we agree with your comments. We will double-check the content of the article and revise it carefully, both the previously mentioned comments and this one.

Round 3

Reviewer 1 Report

Comments and Suggestions for Authors

Still problems of plant taxonomy about the sentence Dryopteris fragrans (L.) Schott is located in Filicales in the taxonomic system and belongs to the Dryopteris genus of the Dryopteris family. It is not possible that genus and family have the same name and the nomenclature rules assign that the name of the family must end in –aceae. Therefore, the sentence must be changed in Dryopteris fragrans (L.) Schott is located in Filicales in the taxonomic system and belongs to the Dryopteris genus of the Dryopteridaceae family. Be careful, Dryopteridaceae not in italic. To my knowledge, Nicotiana tabacum is Nicoltiana tabacum L.

Author Response

Comments 1: Still problems of plant taxonomy about the sentence Dryopteris fragrans (L.) Schott is located in Filicales in the taxonomic system and belongs to the Dryopteris genus of the Dryopteris family. It is not possible that genus and family have the same name and the nomenclature rules assign that the name of the family must end in –aceae. Therefore, the sentence must be changed in Dryopteris fragrans (L.) Schott is located in Filicales in the taxonomic system and belongs to the Dryopteris genus of the Dryopteridaceae family. Be careful, Dryopteridaceae not in italic. To my knowledge, Nicotiana tabacum is Nicoltiana tabacum L.

Response 1: Thank you for pointing this out and we agree with you. It was my mistake that I didn't get the real rules of plant taxonomic nomenclature. Therefore, we have checked the article in detail and revised it carefully according to the international naming rules.
